

# Attenuated atmospheric backscatter profiles measured by the CO₂ Sounder lidar in the 2017 ASCENDS/ABoVE airborne campaign

Xiaoli Sun[1], Paul T. Kolbeck[2], James B. Abshire[1,2], Stephan R. Kawa[1], and Jianping Mao[1,2]

[1]NASA Goddard Space Flight Center, Greenbelt, Maryland 20771, USA
[2]University of Maryland, College Park, Maryland, 20742, USA

*Correspondence to*: Xiaoli Sun (xiaoli.sun-1@nasa.gov)

**Abstract.** A series of attenuated atmospheric backscatter profiles were measured at 1572 nm by the CO₂ Sounder lidar during the eight flights of the 2017 ASCENDS/ABoVE (Active Sensing of CO₂ Emission over Nights, Days, & Seasons mission, Arctic Boreal Vulnerability Experiment) airborne campaign. In
addition to measuring the column average CO₂ mixing ratio from the signals reflected by the ground, the CO₂ Sounder lidar also recorded the height-resolved attenuated atmospheric backscatter profiles beneath the aircraft. We have recently processed these vertical profiles with a 15-m vertical spacing and 1-s integration time along the flight path (~200 m) for all the 2017 flights and have posted the results at NASA Oak Ridge National Laboratory (ORNL) Distributed Active Archive Center (DAAC) for
Biogeochemical Dynamics (Sun et al., 2022), https://doi.org/10.3334/ORNLDAAC/2051. This paper describes the principle of these lidar backscatter profile measurements, their data processing, and estimated signal to noise ratio.

## 1 Introduction

NASA Goddard Space Flight Center (GSFC) developed the pulsed multi-wavelength CO₂ Sounder lidar
for measuring column average CO₂ mixing ratio (XCO₂) near 1572 nm (Abshire et al., 2018) as an airborne demonstrator for NASA's planned Active Sensing of CO₂ Emission over Nights, Days, & Seasons (ASCENDS) space mission (Kawa et al., 2018). The airborne CO₂ Sounder lidar has operated successfully during several campaigns, including in the 2017 Arctic Boreal Vulnerability Experiment (ABoVE) airborne campaign on the NASA DC-8 aircraft. The CO₂ Sounder lidar uses a pulsed laser and
rapidly step-tunes the laser wavelength across the 1572.335 nm CO₂ absorption line. The lidar receiver uses a single photon sensitive HgCdTe avalanche photodiode (APD) (Sun et al., 2017) and an analog to digital convertor (ADC) which records the entire atmospheric backscatter profile as well as the surface reflected signal. The surface returns are used to retrieve the XCO₂ of the atmosphere column travelled by the laser pulse (Sun et al, 2021). Recently, we have processed the airborne atmospheric backscatter
data to provide the atmospheric backscatter profiles from the airplane flight altitude to the surface with 15-m vertical sampling interval for all eight flights of the 2017 campaign. The results have been posted on the repository for NASA Distributed Active Archive Center (DAAC) for Biogeochemical Dynamics (Sun et al., 2022). The backscatter profiles, although not used directly in the XCO₂ retrieval, provide important context for interpreting the retrieved XCO₂ measurements. They can be used to identify



clouds for retrieving partial column $XCO_2$ to cloud tops (Mao et al., 2018). They also enable identifying and profiling smoke plumes from wildfires and assessing their impact on $XCO_2$ (Mao et al. 2021).

This paper describes the data processing of the airborne $CO_2$ Sounder lidar's atmospheric backscatter profiles from the eight flights of the 2017 airborne campaign. We first give a brief description of the lidar and its data structure. We then describe the details of the Level-0 and Level-1
data processing and the signal to noise ratio (SNR) of the profiles.

**2. A brief description of the $CO_2$ Sounder lidar instrument**

**2.1 The airborne $CO_2$ Sounder lidar**

The airborne $CO_2$ Sounder lidar transmits laser pulses toward nadir from the aircraft and detects and records the laser signal backscattered from the atmosphere and the surface. The wavelength of the single
frequency laser is rapidly stepped tuned across the $CO_2$ absorption line centered at 1572.335 nm. The transmitted laser pulse energies are also measured and are used to normalize the received signal to correct for variations in laser power at the different wavelengths. Each laser pulse propagates through the atmosphere column before it is reflected by the ground surface. The lidar receiver digitizes the pulse waveforms from atmospheric backscatter as well as the surface reflection. Figure 1 shows a simplified
instrument block diagram. More details about the airborne $CO_2$ Sounder lidar can be found in Abshire et al. (2018).

The laser emits 1-$\mu$s wide pulses continuously at 10 kHz. The laser wavelength is step-tuned across the $CO_2$ absorption line for 30 pulses, followed by 2 pulses during the period of wavelength rewind. The wavelengths of the 30 laser pulses during a wavelength scan are listed in Table 1. The laser
wavelengths used by the lidar are offset-locked to $CO_2$ absorption line in a gas cell at 40 mbar pressure and 296 K temperature. The absorption line center is at 1572.335 nm according to HITRAN 2012 (Numata et al., 2011, 2012). The offset locking frequency is changed between pulses to step the laser wavelength across the CO2 absorption line. Figure 2 shows the laser wavelengths plotted across the $CO_2$ absorption line computed from a modelled atmosphere. Note that the distribution of laser wavelengths is
slightly offset from the line center of the absorption line due to the difference in pressure between the $CO_2$ in the reference cell and that of the atmosphere being modelled and measured. The laser pulse emission times, the wavelength scan and the data acquisition are all synchronized to the Coordinated Universal Time (UTC) by the on-board computer. The lidar receiver employs a photon-sensitive HgCdTe APD detector (Sun et al., 2017). The detector has a linear analog response and is capable to
record both the time-resolved atmospheric backscatter profiles and the surface reflected signals within its linear dynamic range. The output of lidar detector is digitized continuously with 16-bit resolution at 100 MHz sampling rate.



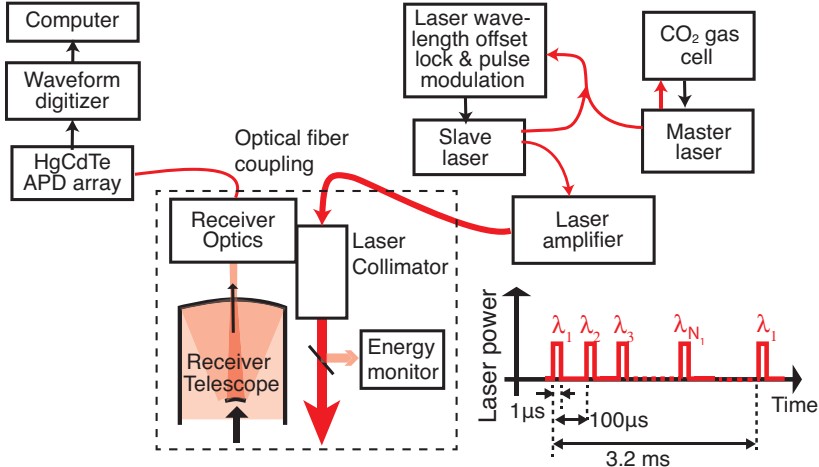

**Figure 1**. A simplified block diagram of the airborne CO₂ Sounder lidar.


**Table 1**. List of laser wavelengths used in the 2017 airborne campaign for the CO₂ Sounder lidar. All wavelengths are given
as the difference of the actual wavelength from the center of the 1572.335 nm CO₂ absorption line in the
75                                                        reference cell.

| Laser pulse number | Frequency and Wavelength difference from the line center of the $CO_2$ cell | | Laser pulse number | Frequency and Wavelength difference from the line center of the $CO_2$ cell | |
|---|---|---|---|---|---|
| | GHz | Picometer | | GHz | Picometer |
| 1 | 12.25 | -101.013 | 16 | -0.25 | 2.062 |
| 2 | 9.50 | -78.388 | 17 | -0.50 | 4.123 |
| 3 | 8.25 | -68.031 | 18 | -0.75 | 6.185 |
| 4 | 6.25 | -51.539 | 19 | -1.00 | 8.247 |
| 5 | 4.25 | -35.047 | 20 | -1.25 | 10.308 |
| 6 | 3.25 | -26.801 | 21 | -1.50 | 12.370 |
| 7 | 2.50 | -20.616 | 22 | -1.75 | 14.432 |
| 8 | 2.00 | -16.493 | 23 | -2.00 | 16.493 |
| 9 | 1.75 | -14.431 | 24 | -2.50 | 20.617 |
| 10 | 1.50 | -12.370 | 25 | -3.25 | 26.802 |
| 11 | 1.25 | -10.308 | 26 | -4.25 | 35.048 |
| 12 | 1.00 | -8.246 | 27 | -6.25 | 51.542 |
| 13 | 0.75 | -6.185 | 28 | -8.25 | 68.037 |
| 14 | 0.50 | -4.123 | 29 | -9.50 | 78.346 |
| 15 | 0.25 | -2.062 | 30 | -12.25 | 101.026 |

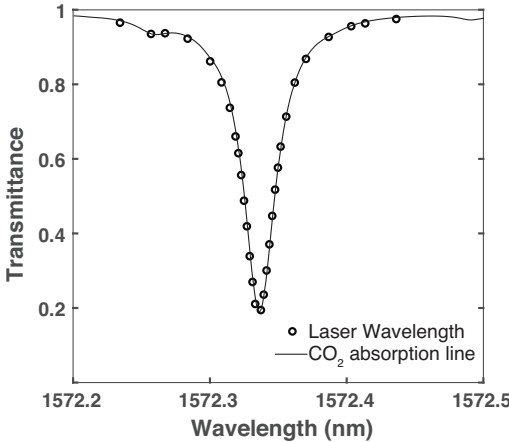

**Figure 2.** The laser wavelengths of the $CO_2$ Sounder lidar overlaid on a typical $CO_2$ absorption line computed for the nadir path using a modelled atmosphere.

## 2.2 Data structure of the airborne $CO_2$ Sounder lidar

The $CO_2$ Sounder lidar operations are synchronized with the UTC time via a GPS receiver. The measured signals are stored into data files at the end of every UTC second. The aircraft housekeeping and the GPS data are also recorded every UTC second. Each lidar data file contains 9 groups of 30 received backscatter profiles, one profile for each wavelength averaged over 32 wavelength scans. Therefore, each group contains 102.4 msec of measurement data (~10 Hz sample rate) and each 1-s data file contains measurement data from $32 \times 9$ wavelength scans. A DC offset of -1.1 V is added to the received waveforms before the ADC to make use of its full dynamic range (±1.25 V). At the 10 kHz pulse rate, the time between transmitted laser pulses is 100 μs. At 100 MHz ADC sampling rate, each waveform segment contains 10,000 ADC sample points. The transmitted laser pulse waveforms are sampled and averaged by another ADC at the same sampling rate for 1,000 points without adding a DC offset. The lidar controller suspends the laser wavelength scan after all the 9 groups of data (total 921.6 ms) are collected and uses the remaining time to generate the timestamp and header for the data file, append the transmitted laser pulse waveforms after the 30th received waveform, and close and save the file. Figure 3 shows a sample plot of recorded data file in the 16-bit signed integer format used by the ADCs.

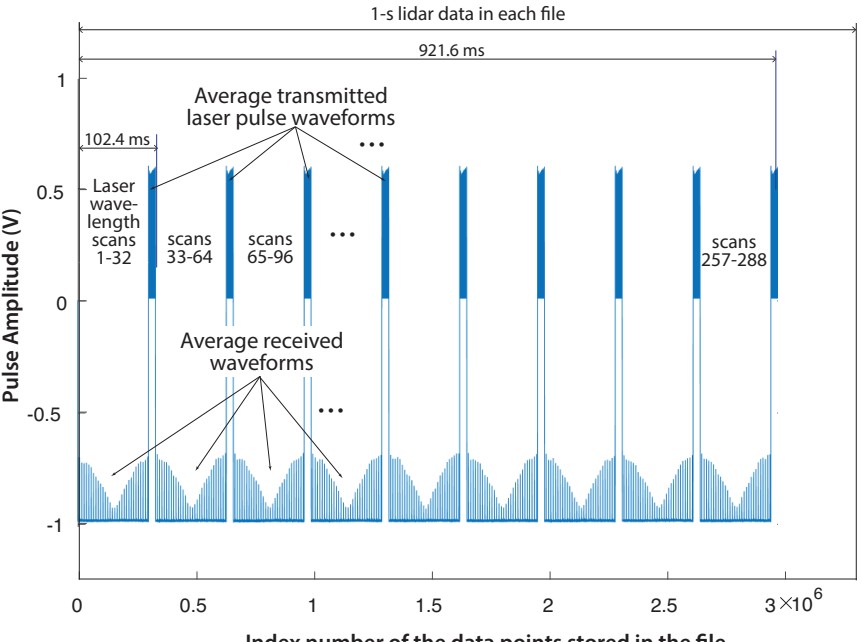

**Figure 3.** Plot of the recorded data file structure for $CO_2$ Sounder lidar. The data were taken on 8/8/2017 23:30:00 (UTC) over the mountains as the aircraft flew toward the Edwards Airforce Base in California at an altitude of 11.2 km. There are nine groups of 30 pulse waveforms, one for each laser wavelength in the scan. Each is averaged over 32 scans. The corresponding waveforms for the 30 transmitted laser pulses, which are sampled at the same rate for 1,000 points each, are appended to the end of the 30th received pulse waveform.

The first 30 individual pulse waveforms from the first group taken on 8/8/2017 at 23:34:00 (UTC) are plotted in Figure 4. The first group of pulse waveforms with relatively low amplitude are from the scatter from the aircraft's nadir window. These are used as the time reference in the laser pulse time-of-flight measurements. The second group of waveforms about 50 µs after the window returns is from thin clouds, and the last set of waveforms are from the ground. Because of the gradual depletion of energy stored in the laser gain media, the laser pulse shape is near rectangular but with its peak power gradually decreases over the 1 µs pulse width period due to the depletion of the laser gain medium. The detector has a slight ringing after the relatively strong ground return, which is omitted in the subsequent signal processing. There is also a small baseline offset in addition to the -1.1 V added before the ADC. The total baseline offset is estimated by averaging the waveform segment before the transmitter pulses and removed in data processing. The 30 pulse waveforms reflected by the window have nearly the same pulse amplitude. The cloud and ground return pulse waveforms show increasing variation in the detected pulse amplitude caused by atmospheric $CO_2$ absorption as the laser wavelengths step through the $CO_2$ absorption line.

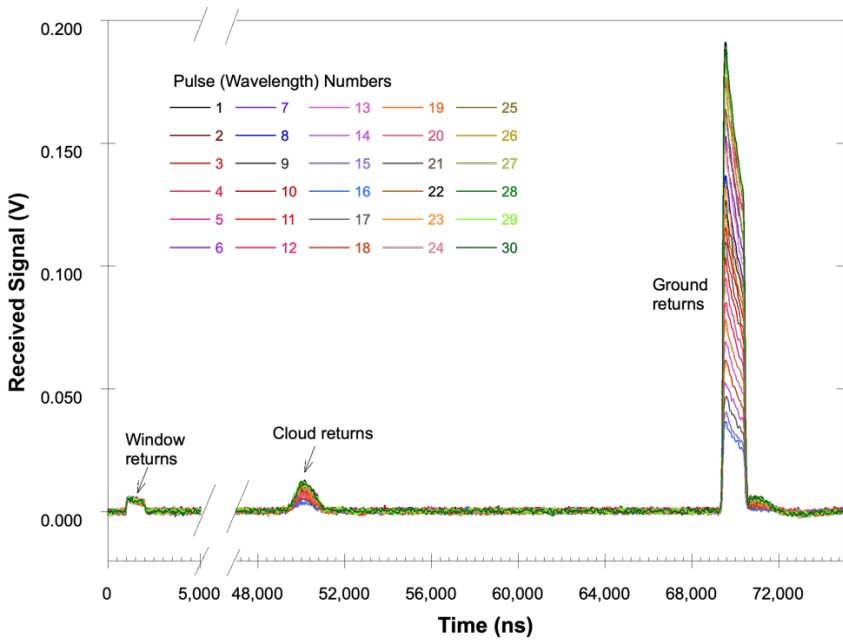

Figure 4. Overlay of the 30 received pulse waveforms recorded by the airborne lidar receiver on 8/8/2017 at 23:34:00 (UTC).

The $XCO_2$ is retrieved from the received laser pulse energies reflected by the surface. At each wavelength the received pulse energy is calculated by first removing the DC offset, then integrating the pulse waveforms reflected by the ground surface and normalizing them by transmitted pulse energies. The relative atmospheric transmissions for each of the laser wavelengths are calculated as the normalized received pulse energies divided by the square of the range measured by the lidar. The surface reflectance is assumed constant during the wavelength scans and so changes in the received pulse energies at different wavelengths are attributed to the absorption by atmospheric $CO_2$. The $XCO_2$ value is retrieved via an algorithm using a linear least-squares fit of the modelled $CO_2$ line shape to the lidar sampled absorption line shape. Details about the signal processing for the $XCO_2$ retrieval can be found in Sun et al. (2021).

The signals used to calculate the atmospheric backscatter profile are contained in the same data files and are obtained by processing the signal waveforms before the ground returns. Some preliminary backscatter profiles in raw ADC units (integers) were reported earlier for this campaign (Allan et al., 2018). We have since then calibrated the backscatter profiles and released results on a public assessable data depository. Details of the data processing and the estimated SNR of the profiles are presented in the remainder of this paper.



**3. Data processing of the atmospheric backscatter profiles from the CO₂ Sounder lidar**

**3.1 Level-0 Data Processing**

The Level-0 data processing converts the raw ADC output in 16-bit integers to the atmospheric backscattered signal in physical unit (V) of the detector output. It also removes all the known artifacts in the data such as baseline DC offsets. To improve the SNR, we averaged the backscatter profiles
measured for laser wavelengths 2, 3, 4, 27, 28, 29 and 30 (7 in total). As shown in Figure 2, these wavelengths undergo little $CO_2$ absorption, so this average may be considered as a backscatter measurement at "off-line" wavelengths.

For each received pulse waveform, the DC offset from the detector is first subtracted by calculating the average detector signal for the time interval that occurs immediately before the return
from aircraft window. Since the pre-window-return region is long after the ground return from the previous laser pulse but before the next one, it is composed primarily of detected solar background, the dark noise from the detector, and the DC offset of the lidar receiver. One component of the DC offset is a constant offset (-1.1 V) added to the detector output before the ADC. The other component is the inherent DC offset from the detector, which can drift slowly over time. It also can change after the
detector is overloaded by excessively strong signals, such as returns from clouds immediately under the aircraft or returns from specular reflection at the ground.

The transmitted pulse energies are calculated from the transmitted pulse waveforms by first subtracting the DC offset and then integrating the signal over the pulse period. They are then used to normalize the corresponding received waveform. The average transmitted pulse energy over the entire
flight is also recorded in the Level-0 data file. A known time delay within the lidar electrical and optical system (corresponding to a range of 26.4 m) is removed next. Following this, received pulse waveforms in the 1-s data file are averaged together to improve the SNR. Finally, the averaged pulse waveform is converted into the units of the detector output (V) by dividing the waveform data by the scale factor of the ADC.

Screening is then performed to eliminate any data with the following abnormalities: the transmitted pulse waveforms are missing in the data file; the ground return pulse is saturated with pulse amplitude above 1.1 V after removing the DC offset; or data are taken while the detector is still recovering from saturation where the estimated detector DC offset falls below 0 V or above 0.5 V after removing the know 1.1 V offset (nominal DC offset from the detector is 0.1 to 0.2 V). The mean and
standard deviation of the waveform within the pre-window region for each averaged waveform are recorded in the Level-0 data files in case further screening is needed. Lidar data that contain cloud returns within 3-km of the aircraft window are also flagged because within this distance the laser beam does not completely overlap the receiver field of view (Allan et al., 2018). The data within this range needs to be further calibrated to account for the losses due to the overlap factor.

The aircraft's navigation data are then merged with the lidar data. The aircraft flight data are obtained from archived airborne campaign housekeeping data gathered during the flight from the on-board GPS receiver and other aircraft instruments. The following parameters are extracted from the archived DC-8 housekeeping data for each second: the UTC time, latitude, longitude, altitude, pitch and roll angles, and the DC-8's radar-measured range to the surface in nadir direction.



185        The effect of off-nadir pointing is corrected next. The lidar is mounted to the aircraft deck and the laser beam is pointed down perpendicularly to the deck. The aircraft flies slightly pitched up during cruise. It also performs pitch and roll maneuvers from time to time, which causes the laser beam to point away from nadir. To correct this effect, the range for the atmospheric backscatter profiles are corrected as if the data are taken along the nadir direction. The range bin size is first multiplied by the cosine of
the combined off-nadir pointing angle and the waveform is divided by the cosine of the off-nadir angle to compensate for the extra signal attenuation by the slant path length. The resultant lidar measurements are resampled by interpolating at 15-m intervals in the nadir direction. Next, all data points in the recorded waveforms above the aircraft altitude and below the ground return are removed. In the cases where there is no discernible ground return, points more than 200 m below the estimated surface
elevation are removed.

       The surface elevation is calculated as the difference between the aircraft altitude from the on-board GPS receiver and the range measured by the lidar after correcting for the effect of off-nadir pointing angle. Missing points in the surface elevation, such as those obscured by clouds, are interpolated linearly between the elevations that border the missing region. The aircraft's latitude,
longitude, altitude, pitch and roll angle are also linearly interpolated during these conditions if the gaps are small (seconds) and the change in measured values before and after the gap is also small.

       Finally, the signal waveforms are vertically smoothed via a boxcar averaging window with the integration time equal to the 1-μs laser pulse width. Since the vertical resolution of the atmospheric backscatter measurements is primarily limited by the laser pulse width, the boxcar averaging has little
effect on the vertical resolution of the backscatter signal, but it reduces higher frequency noise in the detected signal.

### 3.2 Level-1 Data Processing

The Level-1 data processing converts the waveforms from the Level-0 data processing to the attenuated atmospheric backscatter profile in terms of cross section per unit volume per steradian ($m^{-1}$ $sr^{-1}$).
210        The optical signal power collected by the lidar can be written as

$$y(t) = E_{tx} \int_0^{\tau_L} x(\tau)h(t-\tau)d\tau \tag{1}$$

where $E_{tx}$ is the transmitted laser pulse energy in joules, $x(\tau)$ is the normalized laser pulse shape, i.e., $\int_0^{\tau_L} x(t)dt \equiv 1$, $\tau_L$ is the laser pulse width, and $h(t)$ is the impulse response of the atmosphere column, that is, the received signal in response to an infinitely short laser pulse. Here we assume the lidar
detector has a much faster time response than the laser pulse width and the detector's output pulse shape is the same as that of the received optical signal.

       The impulse response of the atmospheric column is related to the volume atmospheric backscatter cross section profile by the lidar equation (Measures, 1984; Reagan et al., 1989)

$$h(t) = \frac{c}{2}\beta[R(t)]\,T_a^2[R(t)]\frac{\eta_r A_{tel}}{R^2(t)}C_1. \tag{2}$$

Here $c$ is the speed of light, $\beta[R]$ is the atmospheric backscatter cross section at range $R$ in units of $m^{-1}$ $sr^{-1}$, $T_a(R)$ is the one-way atmospheric transmission from the lidar to range $R$, $\eta_r$ is the lidar receiver optical transmission, $A_{tel}$ is the collection area of the receiver telescope, $R(t) = c\,t/2$ is the lidar range at time $t$ after the laser pulse emission, and $C_1$ is the lidar detector responsivity in V/W.



Substituting Eq. (2) into (1), the lidar signal from the detector can be written as,

$$s(t) = C_1 \frac{c}{2} \eta_r A_{tel} E_{tx} \int_0^{\tau_L} x(\tau) \frac{\beta[R(t-\tau)] T_a^2[R(t-\tau)]}{R^2(t-\tau)} d\tau. \tag{3}$$

If we assume that the atmospheric backscatter cross section and the atmospheric transmission are approximately constant over the laser pulse interval (1 μs, or 150 m in range), the lidar signal from the detector can be written as,

$$s(t) \approx C_1 \frac{c}{2} \eta_r A_{tel} E_{tx} \frac{\beta[R(t)] T_a^2[R(t)]}{R^2(t)} = C_2 \frac{1}{R^2(t)} \langle \beta[R(t)] T_a^2[R(t)] \rangle \tag{4}$$

where $C_2 = C_1 \eta_r A_{tel} E_{tx} c/2$ is a constant that dependents only on the parameters of the lidar. Therefore, the profile of the attenuated atmospheric backscatter cross section can be calculated from the lidar measurement by

$$\langle \beta[R(t)] T_a^2[R(t)] \rangle = [R^2(t) s(t)] \frac{1}{C_2}. \tag{5}$$

The lidar signal term $s(t)$ in the above equation can be expressed in terms of lidar range by substituting $t = 2R/c$. The attenuated backscatter profile can also be expressed as a function of altitude by subtracting the lidar range from the flight altitude of the aircraft.

For the CO$_2$ Sounder lidar configuration used in the 2017 airborne campaign, the instrument constants are

$$C_1 = \left( \eta_{det} G_{APD} \frac{1}{E_{ph}} e_c \right) Z_{TIA} G_{amp} \eta_{cable}, \quad \text{(V/W)} \tag{6}$$

$$C_2 = C_1 \eta_r A_{tel} E_t \frac{c}{2} = C_1 \eta_r \frac{\pi \phi_{tel}^2}{4} (1 - L_{obs}) E_t \frac{c}{2}. \quad \text{(V m}^3\text{)}. \tag{7}$$

Here $\eta_{det}$ is the detector quantum efficiency, $G_{APD}$ is the APD gain, $E_{ph} = h c/\lambda_{laser}$ is the laser photon energy, $h$ is Planck's constant, $\lambda_{laser}$ is the laser wavelength, $e_c$ is the electron charge, $Z_{TIA}$ is the gain of the preamplifier (a transimpedance amplifier) in V/W, $G_{amp}$ is the post amplifier voltage gain, $\eta_{cable}$ is the electrical cable transmission, $\phi_{tel}$ is the receiver telescope diameter and $L_{obs}$ is the fractional loss of the telescope's collection area caused by its center obscuration.

Table 2 lists the values of these parameters for the 2017 CO$_2$ Sounder lidar. The resultant detector responsivity is $C_1 = 6.39 \times 10^8$ V/W for nominal APD gain, and the instrument constant is $C_2 = 5.13 \times 10^{10}$ V m$^3$.


Table 2. CO$_2$ Sounder Lidar parameters as used in the 2017 Airborne Campaign

| Instrument Parameters | Values |
|---|---|
| Laser pulse energy, $E_t$ | 25 μJ |
| Laser pulse width, $\tau_L$ | 1.0 μs |
| Laser pulse rate | 10 kHz |
| Laser wavelengths, $\lambda_{laser}$ | 1572.2 to 1572.5 nm |
| Telescope diameter, $\phi_{tel}$ | 0.20 m |
| Telescope center obscuration, $L_{obs}$ | 16% |
| Receiver optical transmission, $\eta_r$ | 81.3% |
| Receiver field of view, $\theta_{FOV}$ | 500 μrad |
| Receiver optical filter width, $\Delta\lambda_f$ | 1.4 nm |
| Receiver integration time | 1 s |





| Detector quantum efficiency, $\eta_{det}$ | 69.3%, including the fill factor |
|---|---|
| Detector avalanche gain, $G_{APD}$ | 190 (at 10 V APD bias) |
| Detector excess noise factor, $F_{ex}$ | 1.05 |
| Detector noise equivalent power, $NEP$ | 1.7 fW/Hz$^{1/2}$ |
| Receiver noise bandwidth for atmospheric backscatter calculations, $B_n$ | 0.5 MHz |
| Transimpedance amplifier gain, $Z_{TIA}$ | 320 kV/A |
| Post amplifier voltage gain, $G_{amp}$ | 12.6 |
| Cable transmission, $\eta_{cable}$ | 90% |
| Overall receiver responsivity $(\eta_{det}G_{APD}/hc/\lambda_{laser})$ $Z_{TIA}G_{amp}(1-\eta_{cable})$ | 6.39e8 V/W at 10 V APD bias, decreases by a factor of two for every 1 V reduction in APD bias |

The gain of the lidar detector was adjusted during the flight to prevent saturation from the surface reflected signal as the aircraft changed altitude. The detector gain was changed by a factor of 2
in each step. In data processing, the gain value was determined from the pulse amplitude of the window returns. After filtering out anomalous window returns, which generally resulted from the aircraft flying through clouds, the pulse energies and the centroid of the window returns are calculated and binned for each 1-s data file for each flight. The window return pulse energies are divided into groups separated by a factor of two. The average value from the bins with the APD biased at 10 V is defined as the nominal.
Measurements made with window return pulse amplitude equals to ½ the nominal are assigned to have a gain correction factor of 2 and their backscatter waveforms are multiplied by 2. For those with window return pulse amplitudes equals to ¼ their backscatter waveforms are multiplied by a factor of 4, etc. When the window return pulse energy is an outlier or too weak to determine, the last known valid detector gain value is used.
For each second, the attenuated atmosphere backscatter cross sections are obtained by multiplying the signal waveforms from Level-1 data processing by the square of the lidar range and then by the scaling factor $C_2$, as shown in Eq. (5). Figure 5 shows the atmosphere backscatter profile for the flight on 8 August 2017 along the flight path shown in Figure 6. Figures 7 and 8 show expanded views of the same data as in Figure 6. Figure 9 shows a line plot of the vertical atmospheric backscatter profile
measured at time 23:34:00 (UTC).

The raw lidar signal is digitized at 10-ns (1.5 m) intervals by the ADC to enable meter-level calculations of the lidar range to the surface. However, the vertical resolution of the computed atmospheric backscatter profiles is much wider due to the 1-μs (150 m) laser pulse width. To reduce the data volume, the backscatter profile data are sampled with 100 ns (15 m) bin width. This still
oversamples the backscatter profile but preserves certain temporal features in the data, such as the ground returns and cloud boundaries. The horizontal resolution along the aircraft ground track is the distance travelled by the aircraft in 1-s, or about 200 m. These results from most flights showed the values of attenuated backscatter cross sections within the boundary layer are comparable to those reported by Spinhirne et al. (1997) which were measured at a nearby (1540 nm) laser wavelength.
Similar atmospheric backscatter profiles have been measured with a ground-based 2-μm lidar that used the same type of HgCdTe APD detector (Refaat et al, 2020).



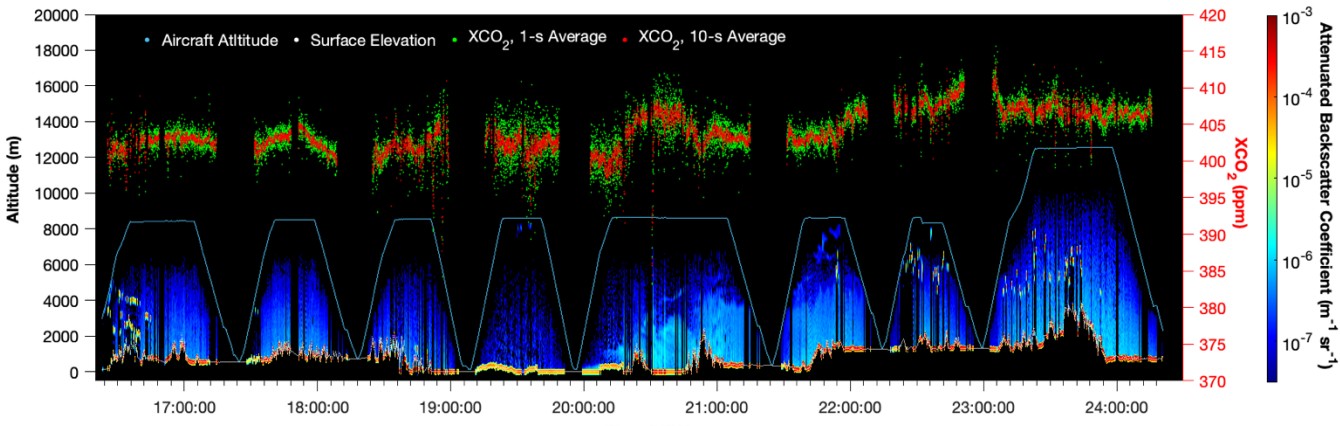

**Figure 5**. Time history of the attenuated atmospheric backscatter profiles measured by the
$CO_2$ Sounder lidar during the flight on 8 August 2017 along with the with the retrieved
values of $XCO_2$.

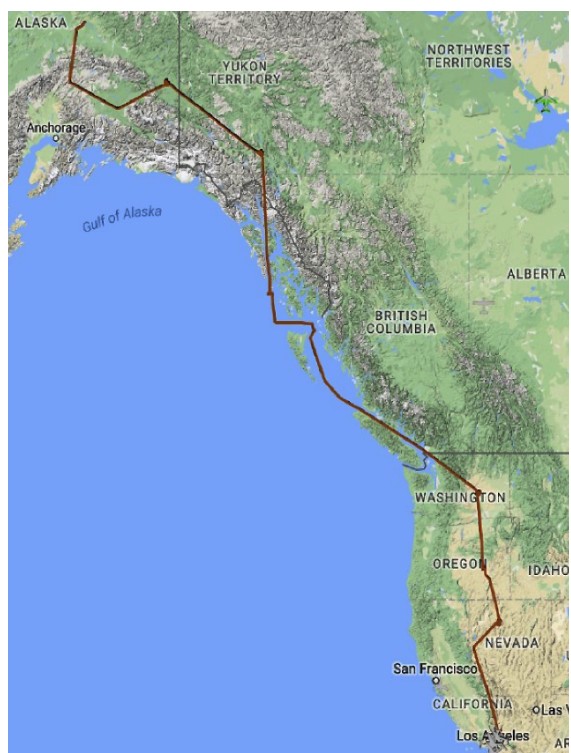

**Figure 6**. The flight path of the airborne $CO_2$ Sounder lidar on 8 August 2017 (plotted via
© Google Maps)

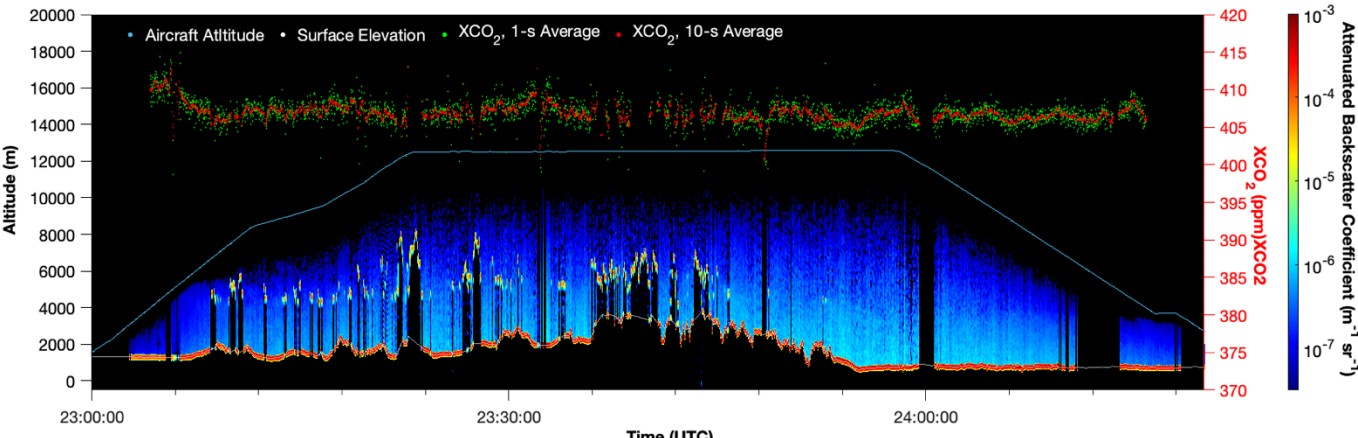

**Figure 7.** Same as Figure 5 but expanded for the last segment from time 23:00:00 (UTC) to the end of the flight.

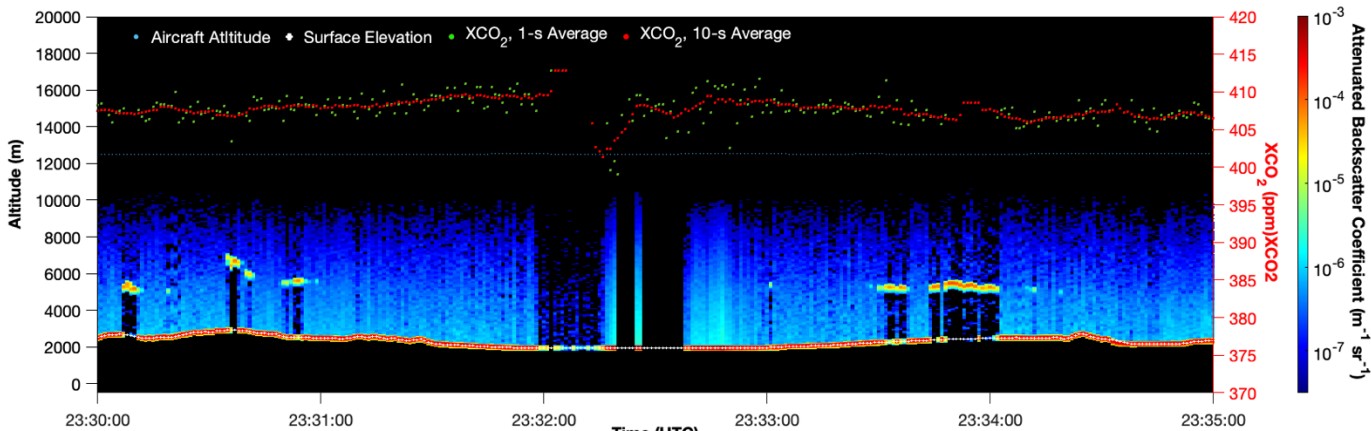

**Figure 8.** Same as Figure 7 but expanded for the 5 minutes between 23:30:00 and 23:35:00 (UTC).

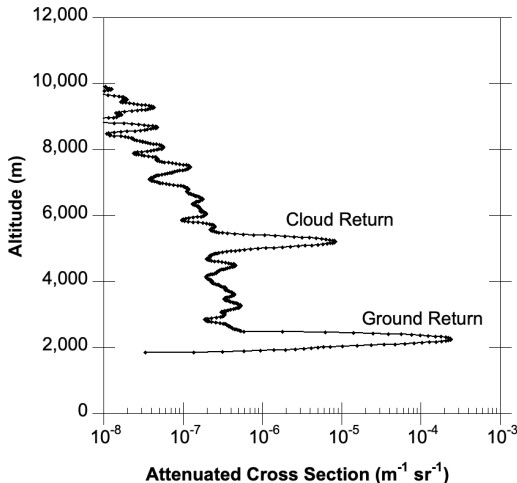

**Figure 9.** Plot of the attenuated backscatter profile and ground return measured on 8 August 2017 at 23:34:00 (UTC) with 15-m vertical sampling interval. The results are from the same data shown in Figure 4 after Level 1 processing.

305

  These backscatter profiles also contain the broadened lidar returns from the ground expressed in terms of backscatter cross section. With the 1.0 μs wide laser pulse, the changes in surface elevation during the receiver integration time usually cause the ground return pulse shape to be broadened to 15 to 20 range bins (1.5 to 2 μs), as shown in Figure 9. These ground return signals at the end of the

310 backscatter profiles can be used to calculate the attenuated surface reflectance by summing the backscatter values over all the range bins containing the ground return and then multiplying the results by π and the bin width (15 m). Figure 10 shows the attenuated surface reflectance as the aircraft approached the Edward Air Force Base on 8 August 2017. Over a mountainous area for the time period from 23:00:00 to 23:40:00 (UTC), the attenuated surface reflectance varied between about 4% when

315 there were clouds and 18% when cloud free. The average attenuated surface reflectance from 0:01:00 to 0:11:00 (UTC) over desert near the end of the flight was 22%. If we estimate the average desert reflectance to be 45% (Kuze et al., 2011), the one-way atmospheric transmission would be 70%, which seems consistent with the sky condition shown on the aerial photograph (Figure 11). When the surface reflectance can be estimated by some other means, the lidar profiles may be used to calculate the

320 extinction to backscatter ratio of thin clouds and aerosols in the path.

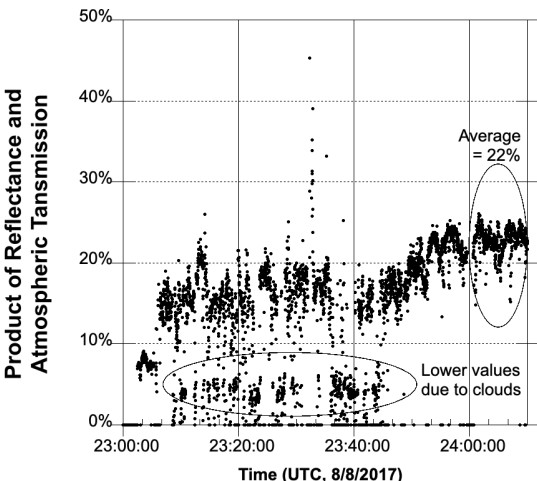

**Figure 10**. Product of the surface reflectance and two-way atmospheric transmission calculated by integrating the ground return of the backscatter profile as the aircraft flew toward the Edwards Air Force Base during 8 August 2017.

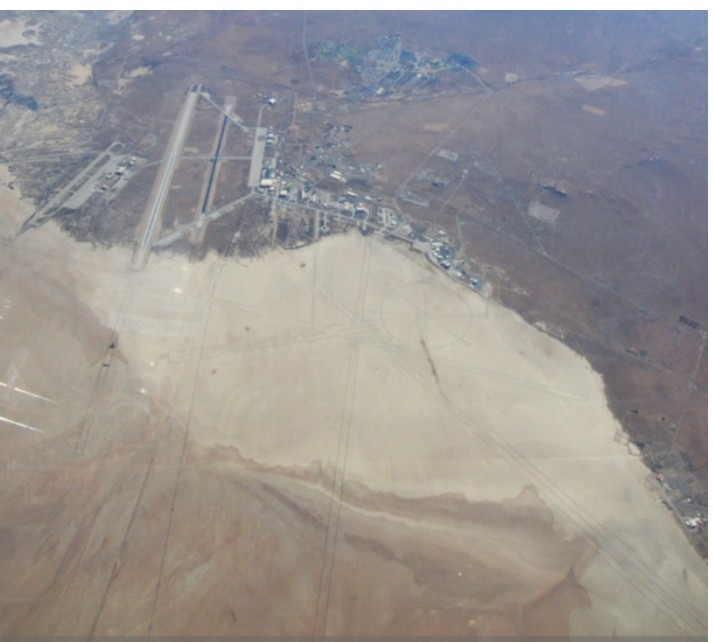

**Figure 11**. Photograph taken from the aircraft to the Edwards Air Force Base on 8 August 2017 24:14:00 (UTC) showing the sky condition for the last segment of data shown in Figure 10.



## 4. Signal to noise ratio calculations

To derive the SNR of the attenuated backscatter profiles we first express the average detected signal
from each laser pulse in a way similar to Eq. (4), but in terms of the rate of detected signal photons as a
function of the lidar range, as

$$\langle \dot{n}_{sig}(R) \rangle \approx \frac{c}{2} \frac{E_{tx}}{E_{ph}} A_{tel} \eta_{det} \eta_r \frac{\langle T_a^2(R) \rangle \langle \beta(R) \rangle}{R^2} \qquad (9)$$

where $R$ is the lidar range.

For each laser pulse, the standard deviation of the rate of the detected signal photons can be
calculated as (Gagliardi and Karp, 1995),

$$\sigma_{noise}(R) = \sqrt{\left\{ F_{ex} \left[ \langle \dot{n}_{sig}(R) \rangle + \langle \dot{n}_b \rangle \right] + \left( NEP \frac{\lambda_{laser}}{hc} \right)^2 \right\} B_n} \qquad (10)$$

where $F_{ex}$ is the detector excess noise factor from the randomness of the APD gain, $\langle \dot{n}_b \rangle$ is the average
detected background photon rate, $NEP$ is the detector noise equivalent power due to the detector dark
noise and the preamplifier noise and $B_n$ is the noise bandwidth of the lidar receiver used for the
atmospheric backscatter calculations. The noise bandwidth is equal to $1/2t_{box}$ with $t_{box}$ the width of the
boxcar averaging window in the signal processing.

For the $CO_2$ Sounder lidar, the lidar range $R$ in Eqs. (9) and (10) consists of a discrete set of lidar
ranges for a series of range bins. Although the attenuated atmospheric backscatter profiles are given in
the 15-m range bin size, the actual profiles are averaged over 150-m vertical layer thickness due to the
effect of the laser pulse width and the boxcar smoothing in the data processing. Therefore, the noise
standard deviation given in Eq. (10) is for the 150-m vertical integration interval. This noise may be
further reduced by averaging data from adjacent 150-m vertical atmospheric layers.

When the lidar views a sunlit surface through a clear sky the average detected background
photon rate is given by

$$\langle \dot{n}_b \rangle = I_s \, \Delta\lambda_f \pi \left( \frac{\theta_{FOV}}{2} \right)^2 \frac{\rho}{\pi} \cdot A_{tel} \, \eta_r \eta_{det} \qquad (11)$$

where $I_s$ is the solar irradiance at the 1572 nm laser wavelength, $\Delta\lambda_f$ is the receiver optical filter
bandwidth, $\theta_{FOV}$ is the receiver field of view diameter, $\rho$ is the diffuse reflectance of the surface, and the
instrument parameter values are listed in Table 2.

For a single laser pulse, the SNR of the attenuated backscatter cross section measurement is the
same as the SNR of the rate of detected signal photons, which is the ratio of Eq. (9) to (10). Therefore,
the SNR of a 150-m vertical bin at range R when averaged over the number of laser pulse measurements
within the 1-s receiver integration time is given by

$$SNR_{av}(R) = \sqrt{N_{ave}} \, \frac{\langle \dot{n}_{sig}(R) \rangle}{\sigma_{noise}(R)} \qquad (12)$$

with $N_{ave}$ the total number of laser pulses measured within the receiver integration time. For the $CO_2$
Sounder lidar with a 1-s receiver integration time in our case $N_{ave} = 32 \times 9 \times 7 = 2016$.

The SNR is proportional to the attenuated backscatter cross section and inversely proportional to
the square of the lidar range. Hence for measurements from an airborne lidar, the SNR is a strong
function of the range to the atmospheric layers being measured. Figure 12 shows the calculated SNR of
the attenuated backscatter cross section vs. lidar range for $10^{-6}$, $10^{-5}$, and $10^{-4}$ m$^{-1}$ sr$^{-1}$ backscatter cross





sections at local noon and night. For example, for a 12-km aircraft altitude above ground surface and a
1-s receiver integration time, the averaged SNR is about 65 for clouds with a backscatter cross section of
$10^{-5}$ m$^{-1}$ sr$^{-1}$ that are 7 km below the aircraft (5 km above surface) during daytime at 45° sun angle.
Therefore, mid-altitude clouds can readily be identified from these attenuated backscatter profiles. The
signals from aerosols at lower altitudes are much weaker. For example, at 10 km lidar range, the
averaged SNR is about 2.9 for aerosols of backscatter cross section of
$10^{-6}$ m$^{-1}$ sr$^{-1}$ at 45° sun angle. This SNR is sufficient to visually identify the planetary boundary layer
from images of the profiles along the flight path, as shown in Figures 7 and 8. It also allows identifying
smoke plumes from wildfires (Mao et al, 2021) that have cross sections > $10^{-6}$ m$^{-1}$ sr$^{-1}$.

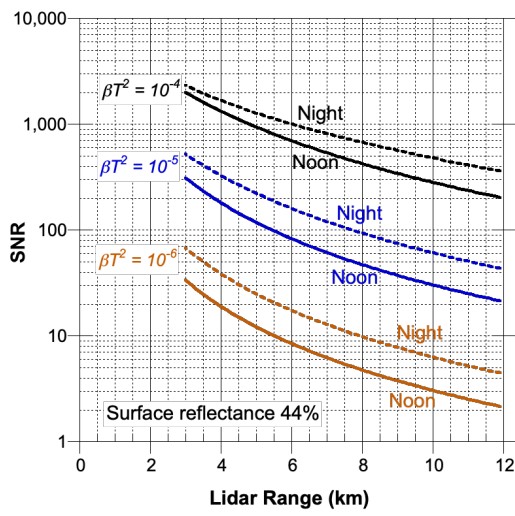


**Figure 12**. Estimated SNR of the attenuated backscatter cross sections obtained from the
$CO_2$ Sounder lidar in the 2017 airborne campaign for 1-s integration times and 150-m
vertical resolution. The SNR of the backscatter profiles may be further improved by
averaging over a longer time intervals and by using thicker atmospheric layers.

**5. Conclusions**

        In addition to the measuring the column average $CO_2$ mixing ratio (XCO$_2$) the NASA GSFC
$CO_2$ Sounder lidar measures the attenuated atmospheric backscatter profiles in the laser beam's path.
We have recently processed the atmospheric backscatter profile data from the 2017 ASCENDS/ABoVE
(Active Sensing of $CO_2$ Emission over Nights, Days, & Seasons mission/Arctic Boreal Vulnerability
Experiment) airborne campaign and produced a new dataset of range-resolved attenuated atmospheric
backscatter cross section at 1572 nm for each of the eight flights. The analysis shows the signal to noise
ratios of the $CO_2$ Sounder lidar backscatter profiles are sufficient to identify clouds, estimate the height
of aerosol layers above the ground, and detect smoke plumes from wildfires. This new data set also
provides additional information to help interpret the retrieved XCO$_2$. These same types of measurements





may be obtained in the future from a space-based lidar that uses the same measurement technique and has a similar power-aperture product and scaled to the spacecraft orbit altitude.

**Appendix**

The following is the flowchart of the data processing software, which was developed using Matlab. The software consists of a main program "backscatterProcessing.m" and a set of functions indicated in blue
text. The software code is available at the same website where the atmospheric backscatter profiles are archived.

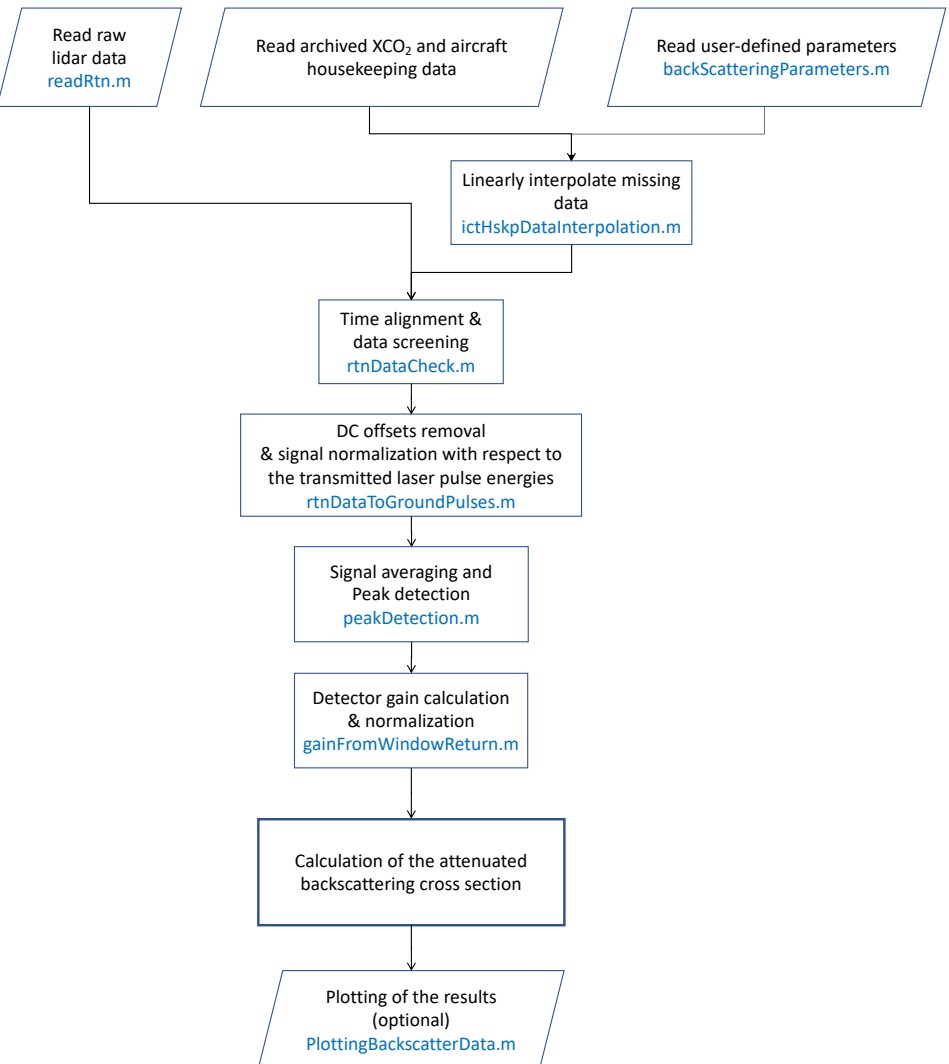

**Figure A1**. Flowchart of the software used to process the data described in this paper.




*Code availability.* The software codes used for the data processing of the atmospheric backscatter profiles described in this paper are posted on the same data depository website as the data.

*Data availability.* The atmospheric backscatter profiles by the $CO_2$ Sounder from the 2017 airborne lidar
measurements are available from the NASA Oak Ridge National Laboratory (ORNL) Distributed Active Archive Center (DAAC) for Biogeochemical Dynamics, https://doi.org/10.3334/ORNLDAAC/2051. NASA Earth Science Data are published under an arrangement equivalent to a CC0 licence. All data are openly shared, without restriction, in accordance with the EOSDIS Data Use Policy posted at https://earthdata.nasa.gov/earth-observation-data/data-use-
policy?_ga=2.94803679.255626558.1646145252-1577112238.1614541217.

*Author contributions.* XS led the data processing and the writing of the manuscript; PTK developed the algorithm and the software to process the data; JBA led the $CO_2$ Sounder lidar team, the instrument development, the 2017 airborne campaign and its data analysis; SRK led the data archiving to the NASA
Airborne Science Data website; and JM processed the 2017 airborne data for $XCO_2$ retrieval; all co-authors participated the writing of the manuscript.

*Competing interests.* The authors declare that they have no conflict of interest.

*Acknowledgements.* We are grateful to Haris Riris, Bill Hasselbrack, Jeffrey Chen and Kenji Numata at NASA Goddard for the development of the $CO_2$ Sounder lidar, supporting the 2017 airborne campaign, and for collecting the lidar measurements. We are particularly grateful to the late Graham Allan for his work on the $CO_2$ Sounder lidar, participating in its airborne campaigns and his early work in analysing the atmospheric backscatter profiles.

*Financial support. This research has been supported by the NASA Earth Sciences Technology Office (ESTO) the NASA ASCENDS Mission pre-formulation program and the 2017 NASA ABoVE campaign.*

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
