# Peer review of "Attenuated atmospheric backscatter profiles measured by the CO2 Sounder lidar in the 2017 ASCENDS/ABoVE airborne campaign"

_Earth System Science Data, 2022_

## Author Response (AR1)

**Manuscript essd-2022-59**

**Attenuated atmospheric backscatter profiles measured by the CO2 Sounder lidar in the 2017 ASCENDS/ABoVE airborne campaign**

Xiaoli Sun1, Paul T. Kolbeck2, James B. Abshire1,2, Stephan R. Kawa1, and Jianping Mao1,2

**22 July 2022**

**Response to Referee #1:**

**General Comments:**

This manuscript presents the data of profile measurements derived from the airborne based CO2 Sounder lidar operating at 1572 nm. Complement to the XCO2 column measurement, this new data set provides the opportunity, to identify clouds, estimate the height of aerosol layers above the ground, and detect smoke plumes from wildfires using a space-based or airborne-based lidar. The method and dataset would benefit retrieval algorithms development for current and future space-based greenhouse gas lidar missions. The presented methods and data are presented clearly and the paper is generally well written and can be accepted after minor revisions.

We thank you for your careful reading of the manuscript and all the corrections and constructive suggestions.

**Specific Comments:**

The flight path includes footprint over water, could you please give statement about the detectability of XCO2 and atmospheric profile over water?

We added three statements at now Lines 363-369. The water (ocean) surface is specular so in most cases scatters less light toward the lidar compared to the ground and consequently produces lower noise in the measurements, as shown in Fig. 5, between Spiral-downs #3 and #4. This improves the SNR of the atmospheric backscatter measurements but can reduce the SNR of XCO2 measurements, depending on wind speed. The effect of surface reflectance on the XCO2 measurements will be addressed in a separate paper about the XCO2 measurements from this airborne campaign.

**What does "the detector gain was changed by a factor of 2 in each step." In line 254 stands for? Please also briefly describe how the gain of the lidar detector was adjusted during the flight.**

We added a sentence about why and how the detector gain was adjusted, now Lines 346-349. We adjusted the detector gain mainly when the aircraft changed its altitude to keep the signal within the receiver linear dynamic range. It was adjusted manually, often during spiral-down maneuvers. The detector gain stayed mostly unchanged when flying at near constant altitudes where the atmospheric backscatter data were obtained.

CO2 absorption lines are difference in pressure and temperature, It is better to also illustrate how the offset locking frequency sample the variable CO2 absorption line in the atmosphere in Figure 2.

The atmospheric backscatter profile measurements were obtained at off-line wavelengths only. The measurements are not very sensitive to small variation in the absorption of the CO2 line due to pressure and temperature. The required precision for atmosphere backscatter profile measurement is much lower than that for XCO2. Therefore, the effects of line shift and broadening are usually buried by other source of noise.

**Technical corrections**

*Line 44: "…records the laser signal backscattered from the atmosphere and the surface." should be "backscattered from the atmosphere and reflected by the surface"*

The correction you recommended has been made.

Line 58: Typo, CO2 should be subscript 2.

The correction has been made.

*Line 95-97: The sentence is too long to follow, and "append" should be "appends", "close and save" should be "closes and saves".*

We rewrote the sentence as several shorter sentences: "The remaining time before the next whole UTC second is used to generate the timestamp and to combine the transmitted laser pulse waveforms into the same file. The pulse waveforms from the 31th and 32nd laser pulses during the wavelength rewind are discarded. The transmitted pulse waveforms are appended to the 30th received waveform."

*Line 103: "nine groups" could be "9 groups", in order to consist with above text (for example line 87).*

The change you suggested has been made.

*Line 210: add a comma after "The optical signal power collected by the lidar can be written as".* The change you suggested has been made.

Lin 262: "equals" should be "equal".

The change you suggested has been made, include another occurrence in the following sentence.

**Response to Referee #2:**

This paper describes a valuable data set of atmospheric backscattering profiles observed with the NASA multi-wavelength airborne CO2 sounder lidar at around the CO2 absorption line at 1572.335 nm. The paper is well written and is recommended for publication after minor corrections.

We thank you for reviewing the manuscript and for your suggestions below.

One thing that is difficult to understand is the second half of the sentence in lines 189-191 "The range bin size is first multiplied by the cosine of the combined off-nadir pointing angle and the waveform is divided by the cosine of the off-nadir angle to compensate for the extra signal attenuation by the slant path length." What is the "extra signal attenuation"? The attenuation cannot be corrected by simply divided by the cosine. If it is about the range square factor, it seems that it should be square of the cosine. ...?

Thank you for pointing this out. We revised the text as follows: "When the laser beam is pointed away from nadir, the laser pulse travels a slant path (a longer distance) before reaching the ground. This effect, if not corrected, would result a longer length profile and a lower ground elevation when plotted with other profiles taken along nadir direction. To correct this effect, the range bin size (15 m) is first multiplied by the cosine of the combined off-nadir pointing angle. This has the effect of compressing the profile and correcting the elevation of the ground return. The resultant is then resampled by interpolating at 15-m intervals in the nadir direction. The attenuated backscatter coefficients in the corrected range bins should also be corrected for the additional attenuation by the atmosphere above the layer. Under uniform layer structure, atmospheric transmission from the aircraft to the given altitude equals  $T(\theta_{nadir}) =$  $[T(0)]^{1/cos(\theta_{nadir})}$  with  $\theta_{nadir}$  the laser beam pointing angle from nadir. The aircraft pitch angle was mostly within 5 degrees during cruise. The roll angle was near zero except during sharp turns or spiral-dows. Total decrease in the two-way transmission is < 5% under this condition. We did not correct this effect in the current data release. Since the combined off-nadir angle is relatively small, it is a relatively small effect for this data set. Additionally, since both the pitch and roll angles are included in the released data, the user may apply this correction if necessary."

**Response to Referee #3:**

This manuscript describes a method to apply the CO2 Sounder Lidar for measuring XCO2 to cloud and aerosol observations, focusing on data processing algorithms and data quality evaluation methods. The main research findings of this manuscript will be particularly important for understanding the relation between XCO2 and aerosols. The method of data processing is well described and will likely become a cited example of how to undertake such tasks. I have commented on the manuscript with several minor corrections, which I believe will improve the readability of the manuscript.

We thank you for your very careful review of the manuscript and providing us suggestions to further improve the manuscript.

1. The authors use "attenuated backscatter cross section" for the same meaning as "attenuated backscatter coefficient." Since the unit of  $\beta$ T2 is m-1 sr-1 (Equation 5), "cross section" may be misleading. In addition, some "attenuated" is omitted. In that case, it would also be misleading as it means the atmospheric backscatter coefficient  $\beta$ .

We have changed "backscatter cross section" to "backscatter coefficients" throughout the revised manuscript. We also added the word "attenuated" where it was omitted.

**2. Lines 149-150.**

The authors are discussing the off-line signal using seven wavelengths except for the No.1 wavelength. You have to clarify why you did not use the No.1 with quite a weak absorption.

We added a statement about why we omitted the No.1 laser pulses. The laser wavelength for the first laser pulse in the wavelength scan were unstable occasionally after the rewind of the lidar's wavelength generator. We did not use this pulse in the XCO2 retrieval. Here we also excluded it for the atmospheric backscatter data.

**3. Figures 5 and 6.**

*I do not understand the relationship between the XCO2 and attenuated backscatter coefficient profiles and the flight path. I suggest you add latitude and longitude to the x-axis of Figure 5.*

To help clarify this aspect for the readers we added text box in Figure 5 to indicate the times for the seven spiral-down maneuvers. We also added the direction of the flight path and a red dot on Figure 6 at the location of each spiral-down. We also added a few sentences in the text to explain the flight path. We hope these will help reader to see the type of ground path (mountain, water, desert, etc.) for each segment of the atmospheric backscatter profiles in Figure 5.

**5. Lines 307-309 (no #4).**

If layers of clouds and aerosols are also broadened to 15 to 20 range bins for the same reason, the effective range resolution of the lidar may not be 10 bins obtained with the data processing. Please include enough information such that an independent investigator would be able to repeat the experiments. Thank you for pointing this out. The reflected laser pulses from flat surface are 1-microsecond wide and span 10 range bins. The box-car filter used to smooth the measurements caused the smoothed waveform to span 20 bins at the base. The text has been corrected.

**6. Lines 376-377.**

I suggest you add a line showing the boundary layer altitude in Figures 7 and 8 to help focus the reader for discussing this effect.

We did not attempt to give the altitude of the boundary layer since it usually belongs to Level-2 data processing. We would like to leave it to the users of the data since the ESSD journal is primarily for description of published data set rather than science outcome from these data.

**7. SNR evaluation of observation results**

You have shown observation results of attenuated backscatter coefficients, XCO2 and surface reflectances. These results are sufficient evidence of the usefulness of the CO2 Sounder Lidar. Unfortunately, the discussion of the SNR is only based on assumptions but remains unconvincing. I suggest you discuss the SNR of clouds and/or smoke plumes using observation data.

We have added a new plot (Fig. 13) to show the mean and ratio of mean to standard deviation from 100 consecutive measurement over a cloud-free stretch on 8 August 2017 23:55:00 (a small segment of data in Fig 7). We also added a few sentences to compare the modeled SNR and the measured SNR approximated by the ratio of the mean to standard deviation under the same assumptions and conditions. The latter is slightly lower because it includes the effect of varying atmospheric condition over that time period. We discussed the SNR for clouds in the text. We dis not describe smoke plume detection since there are already published result (Mao et al., 2020)

8. Line 393.

No data about the detection of smoke plumes are presented in the manuscript hence there is no backing for this conclusion.

We have removed the part of the sentence about smoke plumes in the conclusion.